# Heterogeneity of Rift Valley fever virus transmission potential across livestock hosts, quantified through a model-based analysis of host viral load and vector infection

**Hélène Cecilia**[1¤]*, **Roosmarie Vriens**[2], **Paul J. Wichgers Schreur**[3], **Mariken M. de Wit**[2], **Raphaëlle Métras**[4], **Pauline Ezanno**[1], **Quirine A. ten Bosch**[2]*

**1** INRAE, Oniris, BIOEPAR, Nantes, France, **2** Quantitative Veterinary Epidemiology, Wageningen University and Research, Wageningen, The Netherlands, **3** Wageningen Bioveterinary Research, Lelystad, The Netherlands, **4** Sorbonne Université, INSERM, Institut Pierre Louis d'Epidémiologie et de Santé Publique (IPLESP), Paris, France

¤ Current Address: Department of Biology, New Mexico State University, Las Cruces, NM 88003, United States of America
* helene.cecilia3@gmail.com (HC); quirine.tenbosch@wur.nl (QAtB)

## Abstract

Quantifying the variation of pathogens' life history traits in multiple host systems is crucial to understand their transmission dynamics. It is particularly important for arthropod-borne viruses (arboviruses), which are prone to infecting several species of vertebrate hosts. Here, we focus on how host-pathogen interactions determine the ability of host species to transmit a virus to susceptible vectors upon a potentially infectious contact. Rift Valley fever (RVF) is a viral, vector-borne, zoonotic disease, chosen as a case study. The relative contributions of livestock species to RVFV transmission has not been previously quantified. To estimate their potential to transmit the virus over the course of their infection, we 1) fitted a within-host model to viral RNA and infectious virus measures, obtained daily from infected lambs, calves, and young goats, 2) estimated the relationship between vertebrate host infectious titers and probability to infect mosquitoes, and 3) estimated the net infectiousness of each host species over the duration of their infectious periods, taking into account different survival outcomes for lambs. Our results indicate that the efficiency of viral replication, along with the lifespan of infectious particles, could be sources of heterogeneity between hosts. Given available data on RVFV competent vectors, we found that, for similar infectious titers, infection rates in the *Aedes* genus were on average higher than in the *Culex* genus. Consequently, for *Aedes*-mediated infections, we estimated the net infectiousness of lambs to be 2.93 (median) and 3.65 times higher than that of calves and goats, respectively. In lambs, we estimated the overall infectiousness to be 1.93 times higher in individuals which eventually died from the infection than in those recovering. Beyond infectiousness, the relative contributions of host species to transmission depend on local ecological factors, including relative abundances and vector host-feeding preferences. Quantifying these contributions will ultimately help design efficient, targeted, surveillance and vaccination strategies.

**Data Availability Statement:** Input data and scripts can be accessed at https://git.wur.nl/bosch123/riftvalley_withinhost.

**Funding:** This work was supported by INRAE metaprogram GISA (Integrated Management of Animal Health, project FORESEE), INRAE (HC), Conseil Régional des Pays de la Loire (HC), CIRAD (HC), and the Dutch research council (NWO, One Health PACT project 109986, MdW). The funders had no role in study design, data collection and analysis, decision to publish, or preparation of the manuscript.

**Competing interests:** The authors have declared that no competing interests exist.

## Author summary

Viruses spread by mosquitoes present a major threat to animal and public health worldwide. When these pathogenic viruses can infect multiple species, controlling their spread becomes difficult. Rift Valley fever virus (RVFV) is such a virus. It spreads predominantly among ruminant livestock but can also spill over and cause severe disease in humans. Understanding which of these ruminant species are most important for the transmission of RVFV can help for effective control. One piece of this puzzle is to assess how effective infected animals are at transmitting RVFV to mosquitoes. To answer this question, we combine mathematical models with observations from experimental infections in cattle, sheep, and goats, and model changes in viremia over time within individuals. We then quantify the relationship between hosts' viremia and the probability to infect mosquitoes. In combining these two analyses, we estimate the overall transmission potential of sheep, when in contact with mosquitoes, to be 3 to 5 times higher than that of goats and cattle. Further, sheep that experience a lethal infection have an even larger overall transmission potential. Once applied at the level of populations, with setting-specific herd composition and exposure to mosquitoes, these results will help unravel species' role in RVF outbreaks.

## Introduction

At the beginning of this century, 75% of emerging pathogens in humans were estimated to be zoonotic [1] and 77% of livestock pathogens could be transmitted between different host species [2]. Estimating the relative role different species play in sustaining or amplifying pathogen spread is fundamental for designing control strategies [3–6], yet is hampered by an incomplete understanding of the host(-vector)-pathogen interactions that underlie the spread of these pathogens [7–10].

The potential of a host to contribute to virus transmission is determined by the complex interplay of different factors. For viruses transmitted by arthropod vectors (i.e., arboviruses) these epidemiological interactions are driven both by ecological, population-level factors (i.e., the presence of specific host and vector species and their respective interactions) and the individual-level interactions of the virus with its hosts and vectors. The ability of a host species to infect a susceptible vector upon a potentially infectious contact is determined by the latter. Namely, it derives from i) the viral replication in the host and ii) the ability of a vector to pick up the virus upon blood feeding and subsequently become infected and infectious. While these processes can and have been studied in experimental settings, combining these findings into epidemiologically meaningful parameters is challenging [11–13].

Within-host mathematical models and accompanying inference frameworks have been developed to aid the analysis and interpretation of viral load patterns obtained in controlled infection experiments. Such models provide insights into the biological mechanisms underlying observed patterns [14–18] and how those patterns relate to the clinical expression of the disease [12]. The majority of these modeling efforts are based on viral RNA (or DNA) data, which are indirect measures of infectious virus. Efforts to combine these with infectious virus data (e.g., median tissue culture infectious dose, $TCID_{50}$ or plaque forming units, PFU) have recently emerged for influenza viruses and provide better mechanistic insights into the proportion of particles that are infectious and could contribute to onward transmission [19–24].

Rift Valley fever virus (RVFV) exemplifies the challenges inherent to battling multi-host arboviruses. It was first identified in Kenya, in 1930, after description of an enzootic hepatitis in sheep [25]. The virus has since caused outbreaks throughout the African continent as well

as in the Southwest Indian ocean islands (Comoros archipelago, Madagascar) and the Arabian Peninsula [26]. RVFV mainly affects sheep, goats, and cattle, in which it causes abortion storms and sudden death of newborns [27, 28]. Spillover to humans happens through the handling of infectious animal tissue or by vectorial transmission. While most human infections remain asymptomatic or manifest as a mild illness, symptoms can range from flu-like to hepatitis, encephalitis, retinitis and in the most severe cases, haemorrhagic disease [29]. RVFV vector-borne transmission is mainly mediated by *Aedes* and *Culex* spp. mosquitoes, making its establishment possible in a wide range of ecosystems [30]. While sheep are generally believed to be the most important host species [31–33], efforts fall short of quantifying livestock hosts' relative contribution to RVFV transmission.

Here, we aim to gain more insight into the relative importance of livestock species in RVFV transmission. Using experimental data and mathematical modeling, we derive estimates of hosts' individual potential to transmit RVFV to vectors during their infectious period.

## Results

### Overall approach

We developed a mechanistic compartmental within-host model, representing the infection of target cells and the subsequent production of viral particles, not all of which are infectious (Fig 1). We distinguished the total amount of viral particles produced by infected cells, $V_{tot}$, and the subpart capable of infecting new cells, $V_{inf}$. We fitted this model to time-series of viral RNA (RT-qPCR) and infectious virus (TCID$_{50}$), measured daily in calves (n = 8), lambs (n = 16), and young goats (n = 8) intravenously inoculated with a virulent RVFV strain ([34], Materials and methods). We compared the cell-level basic reproduction number $R_0$ and mean generation time $T_g$, between groups. We quantified the relationship between vertebrate hosts' infectious titers and transmission to mosquitoes using data we extracted through a systematic literature review. Finally, we estimated the net infectiousness of livestock species, a metric proportional to the number of mosquitoes a host would infect over the entire course of its infection.

### Data description

All animals became viremic. In total, 10/16 lambs succumbed to the infection or were euthanized, 3 to 7 days after RVFV inoculation, while others survived until the end of the experiment (2 weeks). All calves and young goats survived until the end of the experiment. Animals reached their maximum RNA levels (average 8.79 log$_{10}$ copies/ml, standard deviation 0.81 log$_{10}$ copies/ml) and infectious titers (average 5.16 log$_{10}$TCID$_{50}$/ml, standard deviation 1.16 log$_{10}$TCID$_{50}$/ml) on day 2 or 3 post-infection.

### Within-host model of RVFV infection

We fitted a within-host model to four datasets, measuring viral RNA and infectious virus in RVFV-infected lambs (surviving; dying), calves, and young goats, using a Bayesian framework (Materials and methods). The model consisted of 10 parameters, 5 of which were held constant (Table 1). We estimated the death rate $\delta$ of infected cells, their total daily production of viral particles $\xi p$, among which $p$ are infectious, the degradation rate $d_{inf}$ of infectious viruses into non-infectious viruses, and the clearance rate $c_h$ of viral particles. Parameter values were then used to calculate the cell-level basic reproduction number $R_0$ and mean generation time $T_g$. Initial conditions were set using elements of the experimental protocol along with a sensitivity analysis (Materials and methods, Table 1). Outputs from the Markov Chain Monte Carlo

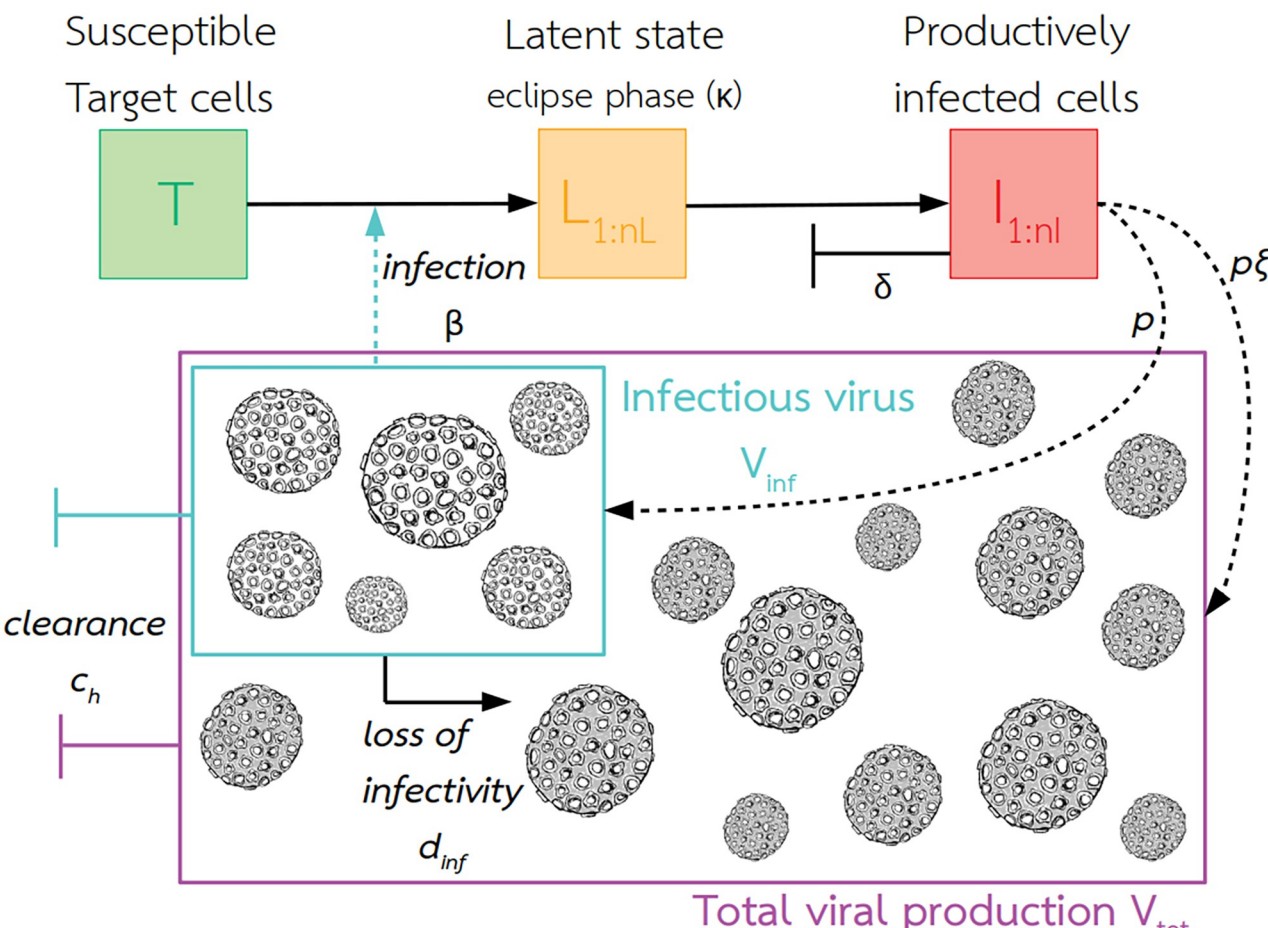

**Fig 1. Graphical representation of the within-host model.** Infectious viruses $V_{inf}$ were fitted to TCID$_{50}$ measures, and total viral production $V_{tot}$ to RT-qPCR measures. The eclipse phase (state $L$) is the period between the infection of a cell by a virus and the presence of mature viruses within the cell. Productively infected cells $I$ are the only ones producing progeny virions. Subscripts in $L$ and $I$ indicate the use of Erlang distributions for the time spent in those states. Target cells are not replenished and only productively infected cells die. Model assumptions, equations and parameter definitions can be found in Materials and methods, Eq (1), and Table 1.

(MCMC) procedure can be found in Section S.1.1 in S1 Text. The fits satisfyingly capture the dynamics present in the data (Fig 2).

The model selection performed highlights different viral load dynamics between livestock species (Deviance Information Criterion (DIC) 1307 *vs* 1186, comparison based on surviving animals as calves and young goats all survived, Fig 2). In particular, the ratio of daily viral RNA over infectious viruses produced ($\xi$) is the highest in the goat group, meaning that the replication process might be less efficient in this species (Table 2). The highest density intervals (HDIs) for this parameter are wide (Table 2), but the posterior distributions remain informative, as knowledge was gained compared to uniform prior distributions (Fig D in S1 Text). In addition, among surviving hosts, the lifespan of infectious particles $(d_{inf} + c_h)^{-1}$ is estimated to be the longest in goats (Table 2). The resulting dynamics show viremia in goats peaks sooner than in calves and in lambs, but with a lower peak value for infectious viruses (Fig 2). Lambs have on average the most infectious viral particles. Model results indicate this could be a result of a slightly higher daily production rate $p$ (Table 2), as well as their initial susceptible cell population, which we estimated to be higher than in other species (Fig A in S1 Text).

**Table 1. Parameters of the within-host model.** Values if fixed, prior range (uniform distribution) if estimated.

| Name | Meaning | Value/Estimated | Reference/Prior |
|---|---|---|---|
| $T_0$ | initial number of susceptible target cells | Fixed within MCMC, estimated *a priori* through likelihood profiles | see Fig A in S1 Text |
| $L_0$ | initial number of cells in latent state | 0 | |
| $I_0$ | initial number of productively infected cells | 0 | |
| $V_{inf,0}$ | initial number of infectious virions | 12.5 for calves, 62.5 for goats, 52.6 for lambs (per ml of plasma, total inoculum per animal being $10^5$) | References for plasma:body weight ratios [35–37] |
| $\beta$ | rate governing infection of target cells by infectious virions | set such as $\beta T_0 = 48$ day$^{-1}$ | assumed |
| $n_L, n_I$ | number of $L$ and $I$ states for the Erlang distributions | 20 | [38, 39] |
| $\kappa^{-1}$ | eclipse phase duration | 1/3 day (8 hours) | P. Wichgers-Schreur personal communication, observed *in vitro* |
| $\delta$ | death rate of productively infected cells | Estimated | [0.1; 10]° day$^{-1}$ |
| $p$ | rate of production of infectious virions | Estimated | [0.2; 3.10$^4$]$^†$ day$^{-1}$ |
| $d_{inf}$ | rate of degradation of infectious virions into non-infectious viral particles | Estimated | [0.1; 10] day$^{-1}$ |
| $c_h$ | host-driven clearance rate | Estimated | [0.1; 10] day$^{-1}$ |
| $\sigma$ | correction factor to convert from infectious virions (plaque forming units) to TCID$_{50}$ | 0.69 | [40] |
| $\xi$ | ratio of total viral particles to infectious virions, as produced by infected cells | Estimated | [1; 1000]$^†$ day$^{-1}$ |

$^†$: these values were applied for each $L$ (respectively $I$) states, so a daily rate per $L$ ($I$) cell (not state) can be obtained by multiplying by $n_L$ ($n_I$)

°: $\delta$ was constrained to be inferior to $\kappa$ and ($c_h + d_{inf}$), as advised by [41].

Characterizing the infectious replication process through the basic reproduction number $R_0$ and generation time $T_g$ (Materials and methods, Eqs (3) and (4)) shows no strong differences between species when comparing surviving individuals (Fig 3). $R_0$ ranges from 8.51 (median; 95% HDI 5.69—14.53) for calves, to 11.47 (median; 95% HDI 7.73—17.68) for lambs. $T_g$ (i.e., the time between infection of a cell and infection of a secondary cell) ranges from 13.48h (median; 95% HDI 12.84h—15.23h) in goats to 14.43h (median; 95% HDI 12.82h—18.31h) in calves.

Among lambs, individuals succumbing to RVF are characterized by higher viral loads, both total and infectious, and a slower decay after the peak is reached (Fig 2). The best model fit is achieved when allowing parameters to vary depending on the survival of the individuals (Fig 2, DIC 928 vs 745), indicating significantly different within-host dynamics depending on clinical outcome. In particular, we estimated that both infected cells and infectious viral particles have prolonged lifespans in dying lambs ($\delta^{-1}$ and ($d_{inf} + c_h$)$^{-1}$ respectively, Table 2). This impacts $R_0$ which is 1.88 times higher (median ratio; 95% HDI 0.84; 3.51) in dying individuals than surviving ones, and $T_g$, which is 1.19 times longer (median ratio; 95% HDI 0.91; 1.65) in dying individuals than surviving ones. Besides, the ratio of daily viral RNA over infectious viruses produced ($\xi$), which does not influence $R_0$, is higher in dying lambs than surviving ones (Table 2).

## Dose-response relationship in RVFV mosquito vectors

Through a systematic review, we identified 9 papers from which data could be extracted to estimate the relationship between vertebrate host infectious titers and associated infection

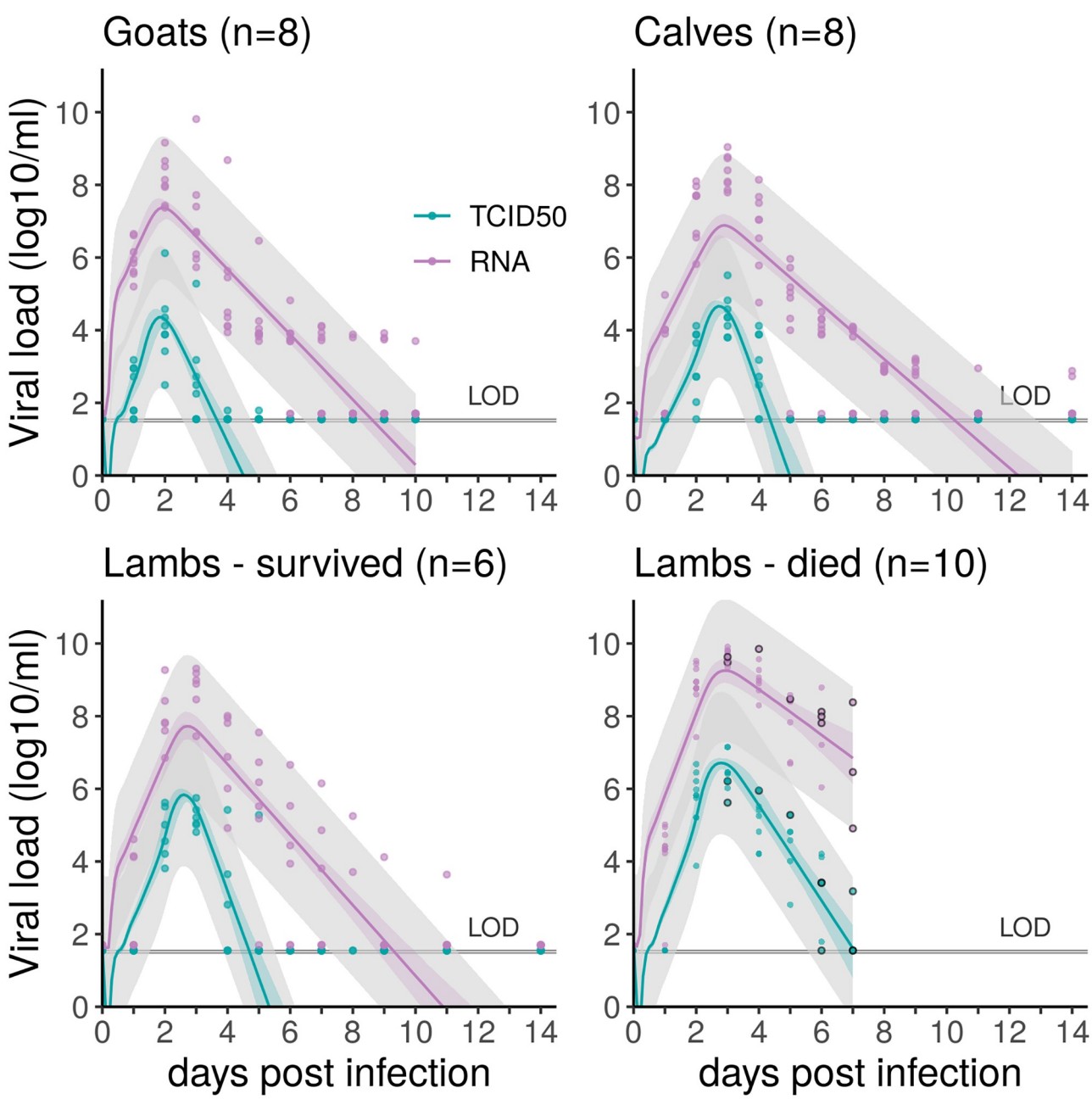

**Fig 2. Data on viral RNA (RT-qPCR) and infectious virus (TCID$_{50}$), in log$_{10}$/ml of plasma, and model fits, for host groups showing significantly different viral dynamics.** Circles are data points. Solid colored lines show the median fit, obtained from 1000 posterior draws. Inner envelopes using the same colors shows the uncertainty from the parameter estimation process (quantiles [2.5–97.5]% of these posterior draws). Outer grey envelopes show the 95% uncertainty bounds associated with the observation process. We assumed this to be normally distributed with a standard deviation of 1 (log10 scale), in line with the sampling error. Purple is for viral RNA and blue for infectious viruses. For lambs which died from RVF, circled points represent individuals' time of death. LOD = limit of detection, 1.55 for TCID$_{50}$, 1.7 for viral RNA (log$_{10}$).

rates in vectors (Materials and methods, Section S.2.1 in S1 Text). Selected experiments were performed with hamster hosts, *Aedes* or *Culex* spp. vectors, using RVFV strain ZH501.

Dose-response curves differ significantly between *Aedes* and *Culex* spp. (Fig 4, Section S.2 in S1 Text). At 5 log$_{10}$ TCID$_{50}$/ml for instance, which most animals could reach or exceed (Fig 2), there is 25% [17; 37] probability to infect an *Aedes* spp. vector and 11% [7; 18] probability

**Table 2. Parameter estimates per host group.** Median of joint posterior distributions (3 chains) and HDI = highest density interval (95%). All parameters are in unit day$^{-1}$, see Materials and methods and Table 1 for detailed definitions. The HDI is built such as every point inside the interval has higher credibility than any point outside the interval [42].

| | Parameter | Estimate: median [HDI] | | | |
|---|---|---|---|---|---|
| | | Goat | Calf | Lamb surv. | Lamb dead |
| $\delta$ | $I$ death rate | 2.61 [1.91; 3.0] | 2.17 [1.30; 3.0] | 2.34 [1.60; 3.0] | 1.52 [0.85; 2.44] |
| $p$ | production of $V_{inf}$ | 20.14 [13.33; 29.40] | 14.98 [12.31; 17.89] | 21.53 [17.09; 26.50] | 25.27 [19.72; 31.00] |
| $\xi$ | ratio $\frac{V_{tot}}{V_{inf}}$ produced | 672.76 [333.96; 999.56] | 75.16 [20.60; 161.05] | 44.33 [9.66; 104.48] | 221.15 [50.57; 510.97] |
| $d_{inf}$ | degradation $V_{inf} \rightarrow V_{tot}$ | 2.10 [1.36; 2.92] | 3.77 [2.17; 6.63] | 3.26 [2.19; 4.48] | 1.60 [0.90; 2.31] |
| $c_h$ | clearance of $V_{inf}$ and $V_{tot}$ | 2.06 [1.88; 2.24] | 1.72 [1.53; 1.92] | 2.24 [1.94; 2.53] | 1.43 [0.87; 2.01] |

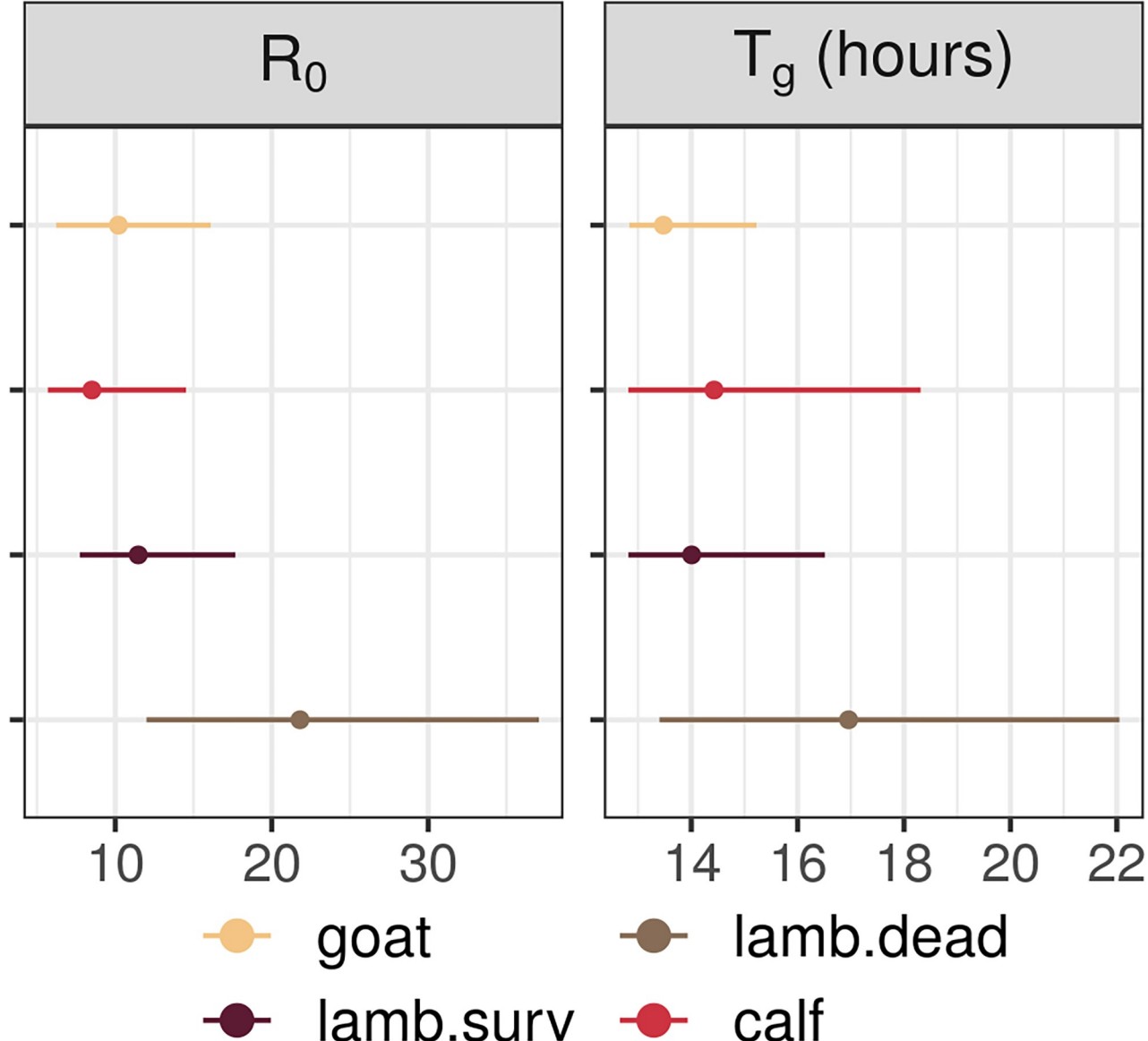

**Fig 3. Outcome measures per host group.** Points are median estimates, lines show highest density intervals, computed from joint posterior distributions (3 chains). Basic reproduction number $R_0$ is computed with Eq (3) and generation time $T_g$ with Eq (4). Note that generation times are constrained in their lower values due to the eclipse phase duration ($\kappa^{-1}$) and rate of virus entry into cells ($\beta$) being fixed (Table 1).

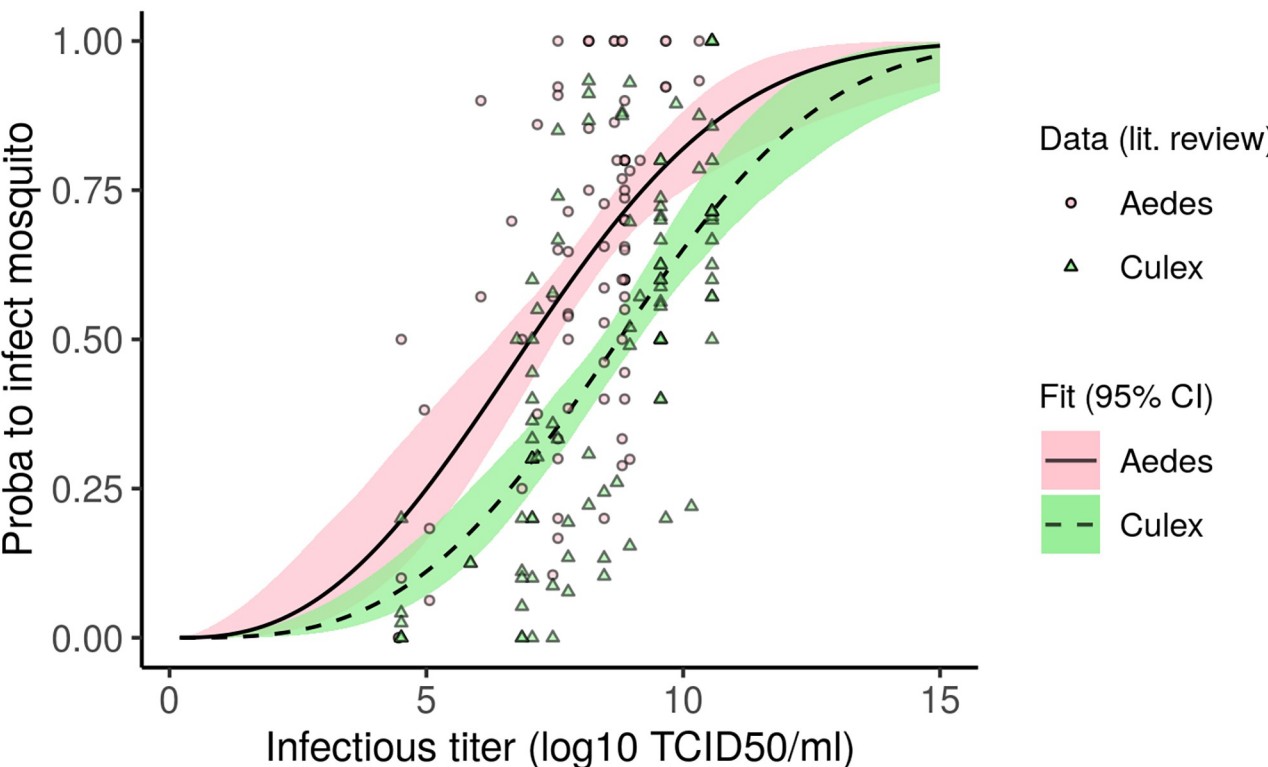

**Fig 4. Dose-response relationships linking host infectious titers to the probability to infect mosquito vectors.** Data retrieved from a systematic review (Materials and methods, Section S.2.1 in S1 Text). Points and triangles show infection rates (presence of RVFV in mosquito bodies, legs excluded) from experiments performed with hamsters with RVFV strain ZH501. Fits were obtained with Eq (S.6) in S1 Text using a betabinomial likelihood to account for overdispersal in the data. Confidence intervals result from 1000 replicate trajectories. Note that infectious titers $>10$ $\log_{10}$ $TCID_{50}$/ml are not to be expected in hosts, but were included to show the full curve.

to infect a *Culex* spp. vector (point estimate and 95% confidence interval, Fig 4). We did not find a significant effect of temperature and number of days post-exposure on infection rates (Sections S.2.2, S.2.3 in S1 Text). The effect of dose is best captured by Eq (S.6) in S1 Text, used by [43], fitted with a betabinomial likelihood accounting for overdispersal in the data (Section S.2.3 in S1 Text). Species-specific curves were estimated for *Aedes vexans*, *Aedes japonicus*, *Culex nigripalpus*, and *Culex tarsalis* (Section S.2.3 and Fig G in S1 Text). While there is intra-genus variability, infection rates in *Aedes vexans* and *Aedes japonicus* are on average higher than in *Culex nigripalpus*, and *Culex tarsalis* at similar host infectious titers (Fig G in S1 Text).

## Net infectiousness of RVFV livestock hosts

Net infectiousness (NI, Eq (5)) varies with both host species and mosquito genus involved (Fig 5). NI is lowest for goats and highest for lambs. The relative differences in NI between host species is stronger when comparing transmission to *Culex* (median ratio lamb:goat 4.79; median ratio lamb:calf 3.75) than to *Aedes* mosquitoes (median ratio lamb:goat 3.65; median ratio lamb:calf 2.93). Every host type studied has the highest NI when bitten by an *Aedes* spp. vector, but the uncertainty around NI estimates decreases when considering *Culex* bites (Fig 5).

Lambs' NI varies with the expected death rate among lambs (Materials and methods, Fig H in S1 Text). Lambs dying from RVF have a higher NI than lambs surviving (Fig H in S1 Text).

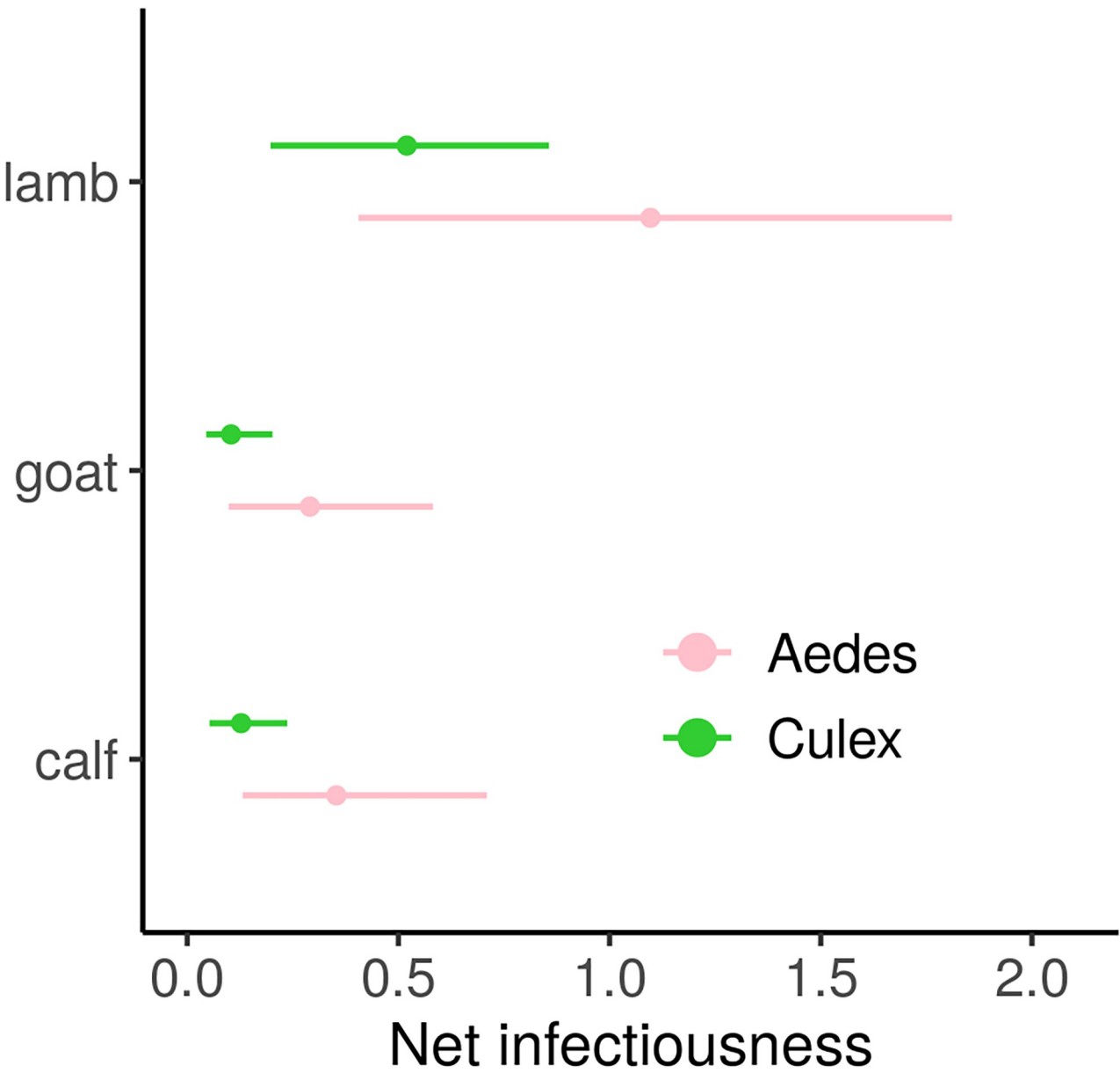

**Fig 5. Net infectiousness of RVFV livestock host species, function of the mosquito genus involved in transmission.** Points are median estimates, lines show highest density intervals, computed using 1000 parameter sets from the within-host model and dose-response curve respective fitting procedures. For lambs, parameters were sampled in the posteriors of both surviving and dying groups, according to the survival rate observed in the original dataset (6/16, Fig H in S1 Text). Time of death also varied according to a Weibull survival model (Materials and methods).

Indeed, dying lambs are more infectious than their surviving counterparts during their whole viremic period, which in 60% of cases can last longer (day 7) than the infectious period of surviving individuals (probability < 1% to infect an *Aedes* or a *Culex* past day 5 post-inoculation). When bitten by an *Aedes* spp. vector, lambs NI ranges from 0.24 to 2.54, increasing by a factor 1.93 (median ratio) from surviving to succumbing individuals. When bitten by a *Culex* spp. vector, lambs NI ranges from 0.12 to 1.36, increasing by a factor 2.22 (median ratio) from surviving to succumbing individuals.

## Discussion

We have presented the results of a data-driven estimation of livestock hosts RVFV transmission potential, providing mechanistic insights into potential sources of heterogeneity between species. Our results demonstrate that sheep are the most infectious livestock hosts, and that virulent infection leading to death reinforces the infectiousness of this species. We also showed that in the current literature, lower infectious doses are needed on average to infect *Aedes* spp. vectors than *Culex* spp.. The framework presented here can be applied to other multi-host arboviruses to estimate transmission potential, a key component of hosts contribution to transmission at large scale.

The suite of experimental data used in our study incorporated the major elements needed for an epidemiologically relevant estimation of hosts' transmission potential. We included both viral RNA and infectious viruses, measured *in vivo*, in natural RVFV hosts. Similar existing models used data coming either from *in vitro* experiments [19, 20, 23, 24, 44], or from model hosts, such as ferrets for influenza [21, 22]. The breeds infected in our dataset, which are dominant breeds from Europe, make our estimates directly relevant for scenarios of RVFV emergence on this continent [45]. A comparison with African breeds is required to know if the relative differences in infectiousness are maintained. Heterogeneity among RVFV strains should also be studied [46]. Performing infection through mosquito bites rather than intravenous injection would ensure a natural course of infection, although the protocol presently used was shown to yield similar viral load dynamics as mosquito-mediated infection [47]. This would further allow for the exploration of the impact of heterogeneity of exposure (i.e., number of infectious bites or infectious titers in vector saliva) on infectiousness. Quantifying more precisely the effect of aging on animals' viral dynamics and pathogenesis is needed to complete our results [48]. Finally, measuring human viral loads, as early as possible post-infection, will be key to complete our understanding of hosts' contribution to RVFV transmission.

Our within-host model is the second developed for RVFV [49], but the first to mechanistically represent the process of viral production from host cells. This enabled an identification of processes driving differences between groups and an increased understanding of the cell-level viral replication process. First, we estimated a less efficient replication in goats, further advocating for the use of infectious virus measures in order not to overestimate transmission potential [50]. Besides, we estimated the lifetime of infectious viral particles and infected cells to be longer in dying lambs than their surviving counterparts, which calls for an exploration of corresponding (immune) mechanisms in future experiments. The uncertainty around parameter estimates remains important, and summarizing parameter estimates into aggregated outcome measures $R_0$ and $T_g$ put those mechanistic differences into perspective. Indeed, once correlations between parameters are taken into account, the replication process is most different between severe and moderate infection within sheep and less so between host species. The model could be refined by incorporating an explicit immune response [51, 52] or taking into account the genomic composition of viral particles [53, 54], but the quantity of information needed (number of timesteps and replicates, inclusion of data on immune responses) could hamper this costly data collection. Alternatively, routinely collected data such as body temperature could constitute an interesting lead to explore time-varying parameters, as a proxy of the immune response. Dedicated modeling would first be needed to determine i) the form of the relationship between temperature and the immune response, most likely with cytokines [55, 56], and ii) which model parameters would be impacted by such an immune response.

By gathering relevant competence studies into a meta-analysis, we quantified the relationship between infectious titers and mosquito RVFV infections. To our knowledge, such dose-response relationship had not been quantified for RVFV. This results in a lack of precision in

between-host RVFV transmission models which usually assume constant infectiousness of hosts over their infectious period. Quantifying how the probability to infect a vector increases with dose will also affect the stochasticity of transmission in small populations (be it emergence or extinction). Dose-response curves have been important for the study of other arboviruses, e.g., for exploring the role of asymptomatic dengue infections [12] or the epidemic potential of *Aedes albopictus* for Zika virus [57]. One important originality of our work was to highlight a higher susceptibility of *Aedes* spp. vectors to RVFV infection compared to *Culex* spp. vectors, at similar infectious titers. Further studies are needed to confirm whether this higher probability of infection is also accompanied by a higher probability of the mosquito becoming infectious itself. This would require the detection of infectious particles in mosquitoes' saliva, which was only performed in 23 out of 185 data points in the present systematic review.

A lot remains unknown about the bottlenecks of arboviruses propagation in mosquitoes [58]. It can depend on species within each genus [59, 60] or even mosquito provenance (field *vs* laboratory-reared, [61–63]), in part because of the role of temperature [64]. Further experiments are needed to know whether a given infectious titer sampled during the increasing or the decreasing phase of viral dynamics would yield the same probability to infect vectors. This comes down to defining what makes a viral particle infectious to host cells *vs* vector cells, and might relate to the efficiency of genome packaging by those cells [54]. Mechanistic modeling will help grasp the complexity of involved processes.

Our study provided key estimates of RVFV livestock hosts' transmission potential. It quantified for the first time the prominent role of sheep, which are 3 to 4 times more infectious than cattle and goats, due to more infectious viruses and a longer infectious period. In addition, fatal infection in sheep does not diminish transmission potential but could rather increase it, based on time of deaths observed in our dataset. This entails that most vulnerable populations, in addition to suffering more deaths, will likely experience larger outbreaks.

Understanding the relationship between infectiousness and pathogen load represents a key challenge to connect modeling scales [65]. We have importantly contributed to deciphering this relationship for Rift Valley fever virus. Combining these results with ecological factors such as vector presence, population dynamics, and trophic preference, as well as human factors, which define the presence of livestock hosts and their mobility, will increase our understanding of RVFV transmission dynamics at large scale. These interacting scales might yield unexpected patterns and reshape the way we design surveillance and control strategies for multi-host arboviruses in general.

## Materials and methods

### Ethics statement

The animal experiment was conducted in accordance with European regulations (EU directive 2010/63/EU) and the Dutch Law on Animal Experiments (Wod, ID number BWBR0003081). Permissions were granted by the Dutch Central Authority for Scientific Procedures on Animals (Permit Number:AVD4010020185564). All experimental protocols were approved by the Animal Ethics Committees of Wageningen Research.

### Experimental design

Data on viral RNA and infectious viruses were obtained from a published study on a candidate RVFV vaccine [34]. Mock vaccinated animals (8 lambs, 8 calves, 8 young goats) were inoculated intravenously with 5 $\log_{10}$ $TCID_{50}$ of strain rRVFV 35/74. Plasma was sampled daily for 10 days in goats, daily for 9 days then every two days until day 14 in calves and lambs. Animals'

age was 2–3 weeks for calves, 8–10 weeks for lambs and goats. The average body weight of animals, used further to calibrate the inoculum per ml of plasma, was 45 kg for lambs, 30 kg for goats, and 80 kg for calves. Animals were purchased from conventional Dutch farms, and the breed was Texel cross for sheep, Saanen for goats, and Holstein-Friesian for cattle [34]. An additional dataset obtained from 8 lambs, following the same protocol, was added.

Viral RNA was isolated with the NucliSENS easyMAG system according the manufacturer's instructions (bioMerieux, France) from 0.5 ml of plasma. Briefly, 5 $\mu$l RNA was used in a RVFV RT-qPCR using the LightCycler one-tube RNA Amplification Kit HybProbe (Roche, Almere, The Netherlands) in combination with a LightCycler 480 real-time PCR system (Roche) and the RVS forward primers (AAAGGAACAATGGACTCTGGTCA), the RVAs (CACTTCTTACTACCATGTCCTCCAAT) reverse primer and a FAM-labelled probe RVP (AAAGCTTTGATATCTCTCAGTGCCCCAA). Virus isolation was performed on RT-qPCR positive samples with a threshold above $10^5$ RNA copies/ml as this has been previously shown to be a cut-off point below which no live virus can be isolated. For the virus isolations, plasma was used. Briefly, BHK-21 cells were seeded at a density of 20,000 cells/well in 96-well plates. Serial dilutions of samples were incubated with the cells for 1.5h before medium replacement. Cytopathic effect was evaluated after 5–7 days post-infection and tissue culture infective dose 50 ($TCID_{50}$) was calculated using the Spearman-Kärber algorithm.

## Within-host model of RVFV infection

Our mechanistic model (Fig 1) is formulated as a set of ordinary differential equations, and is similar to existing within-host models developed for influenza [21, 24]:

$$
\begin{aligned}
\frac{dT}{dt} &= -\beta T V_{inf} \\[2mm]
\frac{dL_1}{dt} &= \beta T V_{inf} - n_L \kappa L_1 \\[2mm]
\frac{dL_i}{dt} &= n_L \kappa (L_{i-1} - L_i), && i = 2, ..., n_L \\[2mm]
\frac{dI_1}{dt} &= n_L \kappa L_{n_L} - n_I \delta I_1 \\[2mm]
\frac{dI_j}{dt} &= n_I \delta (I_{j-1} - I_j), && j = 2, ..., n_I \\[2mm]
\frac{dV_{inf}}{dt} &= p \sum_{j=1}^{n_I} I_j - d_{inf} V_{inf} - c_h V_{inf} - \sigma \beta T V_{inf} \\[2mm]
\frac{dV_{tot}}{dt} &= \xi p \sum_{j=1}^{n_I} I_j - c_h V_{tot}
\end{aligned}
\tag{1}
$$

In this model, infectious viruses $V_{inf}$ infect susceptible target cells $T$ at rate $\beta$. Infected cells first go through a latent state, $L$ (eclipse phase). Then, they become productively infected cells, $I$. These cells produce viral particles $V_{tot}$ at rate $\xi p$, not all of which are infectious ($V_{inf}$ produced at rate $p$). Infectious viruses degrade into non-infectious viruses at rate $d_{inf}$, which does not impact total viral production $V_{tot}$. A similar host clearance rate $c_h$ is applied to both non-infectious and infectious particles.

To achieve realistic distributions of time spent in $L$ and $I$ states, we used Erlang distributions. This means that infected cells go through $n_L$ latent stages and $n_I$ infectious stages, where the time spent in each stage is exponentially distributed. We used $n_L = n_I = 20$, sufficient for the resulting latent and infectious periods to be almost normally or lognormally distributed [38, 39]. The mean of these Erlang distributions are $\kappa^{-1}$ and $\delta^{-1}$, and their variance $\frac{1}{n_L \kappa^2}$ and $\frac{1}{n_I \delta^2}$.

We used a target-cell limited model, meaning that the depletion of target cells is what triggers the viral load peak and subsequent decline. We did not incorporate an explicit immune response. However, as explained by [66], this type of model can be seen as equivalent to assuming a constant effect of the immune response (IR). This IR can act implicitly by limiting the number of cells susceptible to the infection, removing infected cells or viral particles.

We fitted $V_{inf}$ to $TCID_{50}$ measures and $V_{tot}$ to RT-qPCR measures. As $TCID_{50}$ measures the dose needed to induce a cytopathic effect in 50% of the cells, a conversion factor $\sigma$ is needed to express it as a quantity of infectious viruses, usually measured in plaque forming units (PFUs). Here, we set $\sigma = 0.69$ $TCID_{50}$/ml, consistent with 1 ml virus stock having half the number of (PFUs) as $TCID_{50}$ using Poisson sampling [40].

We used a Metropolis Rosenbluth Monte Carlo Markov Chain (MCMC) algorithm to fit our model, implemented in R, using the *odin* package (https://github.com/mrc-ide/odin) to speed up simulations. In our composite log-likelihood $f$ (Eq (2)), we assumed $\log_{10}$ viremia measurements had normally-distributed errors. Below, $\varphi$ and $\phi$ are respectively the probability and cumulative density functions of the normal distribution, and $\epsilon^2$ is taken to be 1 [14]. $D$ is the measure of either viral RNA or infectious viruses (subscript $i$), and $x$ the associated model prediction (either $V_{tot}$ or $V_{inf}$). Measures below the limit of detection (LOD) are considered to be at or below LOD [14]. The final log-likelihood of a given model parametrization is the sum across types of measures $i$, timesteps $j$, and individuals $k$ of the group under consideration.

$$f = \sum_{i,j,k} \log_{10}[\varphi(\log_{10} D_{i,j,k}|\log_{10} x_{i,j,k}, \ \epsilon^2)^{1-c_{i,j,k}} \ \phi(\log_{10} LOD_i|\log_{10} x_{i,j,k}, \ \epsilon^2)^{c_{i,j,k}}]$$

$$c_{i,j,k} = 0 \text{ if } D_{i,j,k} > LOD_i, \text{ else } c_{i,j,k} = 1 \tag{2}$$

$$\log_{10}(LOD_{RNA}) = 1.7$$

$$\log_{10}(LOD_{TCID50}) = 1.55$$

The score $f$ obtained at each iteration was used by the algorithm to determine if a parameter set should be accepted. At each iteration, parameters were simultaneously sampled using normal distributions centered around their last accepted value, with a standard deviation specific to each parameter. To obtain acceptance rates between 10% and 45% (the optimal acceptance rate being 23.4% as shown by [67]) for each parameter, we used a custom function which determines appropriate standard deviations for their sampling. Fixed and estimated parameters can be found in Table 1, chosen in agreement with identifiability analyses of similar models [66, 68]. Priors represent the probability distribution of possible parameter values, based on prior knowledge. We used uniform distributions, with bounds intended to allow a wide exploration of parameter values while being biologically realistic.

Our fitting procedure worked as follows: for each dataset to fit, we ran small chains (10,000 iterations, 5,000 burn-in period) fixing $T_0$ at different values spread across [3;6.5] $\log_{10}$/ml plasma. The best $T_0$ value was then assessed through maximum log-likelihood profiles (Fig A in S1 Text) and kept for longer chains. Three long chains were run (100,000 iterations, 20,000 burn-in) for each dataset. The Gelman Rubin diagnostic test was used to assess common

convergence of the chains (Fig C in S1 Text). Correlation between estimated parameters was assessed (Fig E in S1 Text).

To determine whether viral load dynamics $V(t)$ differ between livestock host groups, we ran the inference procedure in two distinct ways: treating these groups as equal (aggregating datasets) or different (fitting done for each dataset separately, Section S.1.1 in S1 Text). The resulting joint posterior distributions were used to compute the Deviance Information Criterion (DIC) of these models and select those with the smallest DIC (Section S.1.1 in S1 Text). We did not attempt to find differences between individuals of a given group.

To characterize the replication process at the beginning of the infection, we computed two outcome measures from the parameters of our model. The basic reproduction number $R_0$ (Eq (3), [24, 69]) is defined as the average number of new infected cells produced by one infected cell introduced into an entirely susceptible target-cell population. The generation time $T_g$ (Eq (4), [24, 70, 71]) is the average time between the infection of a cell and the infection of a secondary cell, again in an entirely susceptible target cell population. The formula for $T_g$ was adapted to a model using Erlang distributions (for time spent in $L$ and $I$ states). How it changes compared to $T_g$ computed for models with exponential distributions is explained in Section S.1.2 in S1 Text.

$$R_0 = \frac{\beta T_0 p}{\delta(c_h + d_{inf} + \sigma \beta T_0)} \tag{3}$$

$$T_g = \frac{1}{\kappa} + \frac{n_I + 1}{2n_I} \cdot \frac{1}{\delta} + \frac{1}{c_h + d_{inf} + \sigma \beta T_0} \tag{4}$$

## Dose-response relationship in RVFV mosquito vectors

A systematic review of the literature was performed to study $F(V)$, the relationship between a vertebrate host RVFV infectious titer and the associated probability to infect a mosquito upon its bite (Section S.2.1 in S1 Text). We limited our quantitative analysis to experiments performed with *Aedes* and *Culex* spp., with strain ZH501, on hamsters (Section S.2.1 in S1 Text). This corresponded to 185 data points from 9 papers.

To assess the impact of the diversity of protocols from which the data originated, we tested the effect of temperature, and number of days between mosquito feeding and dissection, in addition to dose ($\log_{10}$ infectious titer) on infection rates (presence of RVFV in the body of mosquitoes, legs excluded, Sections S.2.2, S.2.3 in S1 Text). For that we used a logistic function (Eq (S.5) in S1 Text), fitted with a binomial and a beta-binomial likelihood, the latter to account for overdispersal in the data (Section S.2.3 in S1 Text).

We used Akaike Information Criterion (AIC) to compare model fit of different functional forms (Section S.2.3 in S1 Text). Best fitting functions were then used to explore differences between and within genera (Table A and Fig G in S1 Text).

## Net infectiousness of RVFV livestock hosts

We define net infectiousness (NI) as the integral of an infectiousness curve over time (Eq (5))

$$NI_{vect,host} = \int F_{vect}(V_{host}(t))dt \tag{5}$$

NI combines the dose-response relationship in vectors $F_{vect}(V)$ with infectious virus dynamics in hosts $V_{host}(t)$. As such, it must incorporate the uncertainty from both estimations. This was

done by sampling 1000 parameter sets from $F_{vect}(V)$ and $V_{host}(t)$ respective fitting procedures. For lambs, a draw in a Bernoulli distribution first determined whether the viral load dynamics should be of a surviving or dying type. In the latter case, a time of death was sampled in a Weibull survival model fitted to death times present in our dataset, and determined the end of the viral load curve. Finally, a sensitivity analysis explored how the survival rate (probability of the Bernoulli sampling) in the lamb population impacts the average NI of lambs.

This quantity NI is proportional to the expected number of mosquitoes infected by a host over the entire course of its infection, assuming that biting occurs at a constant rate over this period. By extension, the NI ratio of two host categories is identical to the ratio of the expected number of mosquitoes infected by those two types of hosts, assuming bites to be equally distributed over both species. In the present study, NI was also vector-specific.

## Supporting information

**S1 Text. Supplemental details on results. Fig A:** Likelihood profiles to estimate $T_0$. **Fig B:** Trace plots of selected models. **Fig C:** Gelman diagnostic plots, per parameter, for selected models. **Fig D:** Joint posterior distributions of parameters per selected model. **Fig E:** Pairwise correlation between estimated parameters, for each group. **Fig F:** Distribution of temperature, days post-exposure, and infectious titers, in experimental data retrieved from the systematic review, for *Aedes* and *Culex* spp. vectors. **Fig G:** Species-specific dose-response curves. **Fig H:** Net infectiousness of an average lamb in relation with the expected survival rate in the population, for transmission to *Aedes* and *Culex* spp. vectors. **Table A:** Number of datapoints available per vector species, retrieved from the systematic review.
(PDF)

## Acknowledgments

We thank Jeroen Kortekaas for providing the data necessary for our computations, and for stimulating discussions. We thank Hannah Clapham, Sander Koenraadt, Vincent Raquin, Frederick Arnaud, and Maxime Ratinier for fruitful discussions.

## Author Contributions

**Conceptualization:** Hélène Cecilia, Paul J. Wichgers Schreur, Raphaëlle Métras, Pauline Ezanno, Quirine A. ten Bosch.

**Data curation:** Hélène Cecilia, Roosmarie Vriens.

**Formal analysis:** Hélène Cecilia, Quirine A. ten Bosch.

**Investigation:** Hélène Cecilia, Roosmarie Vriens, Raphaëlle Métras, Pauline Ezanno, Quirine A. ten Bosch.

**Methodology:** Hélène Cecilia, Roosmarie Vriens, Paul J. Wichgers Schreur, Raphaëlle Métras, Pauline Ezanno, Quirine A. ten Bosch.

**Resources:** Paul J. Wichgers Schreur.

**Software:** Hélène Cecilia, Quirine A. ten Bosch.

**Supervision:** Mariken M. de Wit, Raphaëlle Métras, Pauline Ezanno, Quirine A. ten Bosch.

**Validation:** Hélène Cecilia, Paul J. Wichgers Schreur, Raphaëlle Métras, Pauline Ezanno, Quirine A. ten Bosch.

**Visualization:** Hélène Cecilia.

**Writing – original draft:** Hélène Cecilia, Raphaëlle Métras, Pauline Ezanno, Quirine A. ten Bosch.

**Writing – review & editing:** Hélène Cecilia, Roosmarie Vriens, Paul J. Wichgers Schreur, Mariken M. de Wit, Raphaëlle Métras, Pauline Ezanno, Quirine A. ten Bosch.

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
