## [Decision Letter · Decision Letter 0]

28 Feb 2022

Dear Dr. Cecilia,

Thank you very much for submitting your manuscript "Heterogeneity of Rift Valley fever virus transmission potential across livestock hosts, quantified through a model-based analysis of host viral load and vector infection" for consideration at PLOS Computational Biology. As with all papers reviewed by the journal, your manuscript was reviewed by members of the editorial board and by several independent reviewers. The reviewers appreciated the attention to an important topic. Based on the reviews, we are likely to accept this manuscript for publication, providing that you modify the manuscript according to the review recommendations.

Both reviewers have provided specific feedback which should be addressed at resubmission. In addition, the authors mention that the log-likelihood was "similar to Clapham et al (2014)" but don't seem to specify it explicitly in the main text or SI. Please give the likelihood in the methods section.

I would also like to thank the authors for referring to the "Metropolis Rosenbluth MCMC" to acknowledge the Rosenbluths' contributions.

Sincerely,

Kat S Rock

Guest Editor

PLOS Computational Biology

Nina Fefferman

Deputy Editor

PLOS Computational Biology

[LINK]

Both reviewers have provided specific feedback which should be addressed at resubmission. In addition, the authors mention that the log-likelihood was "similar to Clapham et al (2014)" but don't seem to specify it explicitly in the main text or SI. Please give the likelihood in the methods section.

I would also like to thank the authors for referring to the "Metropolis Rosenbluth MCMC" to acknowledge the Rosenbluths' contributions.

Reviewer's Responses to Questions

**Comments to the Authors:**

Reviewer #1: The authors have created a nice within-host model that allows them to evaluate variability between hosts in infectious period, and likelihood of infecting a mosquito. My concerns primarily have to do with better documenting the biological methods.

It appears that the data driving the estimates used in this model were obtained from a study of vaccinations. The authors cite the article (Safety and efficacy of four segmented RVFV in young sheep, goats, cattle) for all of their biological methods. I would suggest a brief description in these methods as well - so that a reader at least understands the study design (like fiigure 4 in the citation) and how TCID/viral burden (methods sections in the citation were titled RTqPCR and viral isolation) was calculated since those are the parameters that are important in their model.

As a note - the citations - Wichgers 2020a and 2020b appear to be the same citation. Was this accidental? Was one supposed to be a different citation?

Additionally the authors rightly mention in their discussion that taking into account host immunity would be very important in future iterations. I understand the authors may not want to alter their model for this publication but they do have two very easy to use parameters in the study design if i read the citation correctly - neutralizing antibody titers and body temperature. It would be interesting to specifically mention how they might incorporate those in the discussion.

And lastly it appears the work was done on 1 breed of goat, 1 of cattle and 1 of sheep. And each of those species was of approximately different age. Do the authors think that other breeds of the same species would have any alterations? They should at least mention that they can't be sure that the estimates for sheep would be relevant to all sheep breeds, goat breeds, cattle breeds. Some breeds are bred to be disease resistant and so might have altered estimates. Additionally the authors should mention that the animals were of different ages and clarify whether this is important in their conclusions at all?

Reviewer #2: The study investigates the within-host infection dynamics and their epidemiological consequences of Rift Valley fever virus with an objective to understand the relative importance of livestock species in disease transmission. By fitting a mathematical model of within-host infection to experimental data from relevant livestock species, the authors compare the basic reproductive number and generation time between them. The authors then carried out a literature review to parameterise the relationship between infection load and per capita transmissibility. I find the study to be comprehensive and methodologically sound. The manuscript is concise and well written. Below I list my minor comments.

Given the audience of PLOS Comp Biol, I believe that the abstract should start off a little more general (i.e., the first sentence should introduce the concept, not the studied virus). I think the authors already do a great job at this in their Introduction.

I found the structure of the paper a little strange, with a subsection called “2.2. Data” appearing under the Results section. I recommend merging “2.2. Data” with “4.1. Data”.

Figure 2. The error bands capture the uncertainty from posterior distributions, but ignore the process of sampling. I recommend instead plotting predictive intervals that reflect both the posterior distributions and sampling errors of the data generating process (which is stated to be normally distributed in the Methods).

On page 5, the authors state that “highest density intervals (HDIs) for this parameter are wide (Table 1), but the posterior distributions remain informative (Figure S.4).” But it’s not clear what is meant by “the posterior distributions remain informative”. It doesn’t appear to be a claim about the relative information between the prior and observations as I could not find an assessment for example of posterior contraction in the paper. (sensu https://betanalpha.github.io/assets/case_studies/principled_bayesian_workflow.html).

As within-host models are typically over-parameterised, the reader may benefit from understanding how the within-host parameters related and how much data can inform them. Pairwise plots of posterior draws are typically shown for visualising parameter collinearity.

Figure 4. Posterior draws are falsely labelled as predictive intervals. I recommend calculating the quantiles. Please refer to my comment above in relation to Figure 2.

I found a mixed-use of British and American spelling throughout.

**Have the authors made all data and (if applicable) computational code underlying the findings in their manuscript fully available?**

Reviewer #1: Yes

Reviewer #2: Yes

PLOS authors have the option to publish the peer review history of their article (what does this mean?). If published, this will include your full peer review and any attached files.

Reviewer #1: No

Reviewer #2: No

Figure Files:

Data Requirements:

Reproducibility:

References:

---

## [Decision Letter · Decision Letter 1]

16 Jun 2022

Dear Dr. Cecilia,

We are pleased to inform you that your manuscript 'Heterogeneity of Rift Valley fever virus transmission potential across livestock hosts, quantified through a model-based analysis of host viral load and vector infection' has been provisionally accepted for publication in PLOS Computational Biology.

Best regards,

Kat S Rock

Guest Editor

PLOS Computational Biology

Nina Fefferman

Deputy Editor

PLOS Computational Biology

Jason A. Papin

Editor-in-Chief

PLOS Computational Biology

Feilim Mac Gabhann

Editor-in-Chief

PLOS Computational Biology

Reviewer's Responses to Questions

**Comments to the Authors:**

Reviewer #1: The authors have incorporated my comments from the previous review and I support publication.

Reviewer #2: The authors have sufficiently answered my queries.

**Have the authors made all data and (if applicable) computational code underlying the findings in their manuscript fully available?**

Reviewer #1: Yes

Reviewer #2: Yes

PLOS authors have the option to publish the peer review history of their article (what does this mean?). If published, this will include your full peer review and any attached files.

Reviewer #1: No

Reviewer #2: No

---

## [Editor Report · Acceptance letter]

10 Jul 2022

PCOMPBIOL-D-21-02043R1 

Heterogeneity of Rift Valley fever virus transmission potential across livestock hosts, quantified through a model-based analysis of host viral load and vector infection

Dear Dr Cecilia,

I am pleased to inform you that your manuscript has been formally accepted for publication in PLOS Computational Biology. Your manuscript is now with our production department and you will be notified of the publication date in due course.

With kind regards,

Zsofi Zombor
